# Efficacy of Topical Application of a Skin Moisturizer Containing Pseudo-Ceramide and a Eucalyptus Leaf Extract on Atopic Dermatitis: A Review

**DOI:** 10.3390/jcm13061749

**Published:** 2024-03-18

**Authors:** Yutaka Takagi

**Affiliations:** Faculty of Pharmaceutical Sciences, Josai University, Saitama 350-0295, Japan; ytakagi@josai.ac.jp

**Keywords:** ceramides, barrier function, water holding capacity, sweating, eucalyptus leaf extract (ELE), pseudo-ceramide, SLE66

## Abstract

Atopic dermatitis (AD) is a chronic inflammatory skin disease associated with pruritus, an impaired cutaneous barrier function and a disrupted water holding capacity. Levels of ceramides, which are major components of intercellular lipids and are crucial for their functions, are decreased in the stratum corneum of patients with AD. Treatments to increase ceramide levels are effective for AD care. Synthetic pseudo-ceramide (cetyl PG hydroxyethyl palmitamide (SLE66)), which has a structure developed via molecular designs, and a eucalyptus leaf extract (ELE) enhance ceramide synthesis in the epidermis. The topical application of a skin moisturizer containing SLE66 and ELE improves the barrier functions and water holding capacity of AD skin accompanied by an improvement in skin symptoms. This is a multifaceted review that summarizes the efficacy of the topical application of a skin moisturizer containing SLE66 and ELE on atopic dermatitis.

## 1. Introduction

Atopic dermatitis (AD) is a chronic inflammatory skin disease associated with pruritus. Different etiologies are involved in the pathogenesis of AD, such as skin hypersensitivity, abnormalities of the epidermis, especially the horny cell layer [1,2,3,4], inflammation [5,6], Th2-dominant immune phenotype reaction, and pruritus [7,8,9]. Thus, there are some medical treatments to reduce these skin symptoms. Controlling inflammation is one of the main treatments. Topical corticosteroids, topical calcineurin inhibitor, and biologics are used as drug treatments. The efficacy and safety of topical corticosteroids have been examined in many clinical studies [10] and used as topical drug therapy for AD for over 60 years [11]. Topical calcineurin inhibitors are strongly recommended as alternatives to topical corticosteroids in AD treatment, particularly when there is concern about steroid-induced adverse effects. The biologics prevent the signaling pathway contributing to Th-2 immune responses in AD.

Topical application of emollients is also important for the treatment of AD [12,13,14]. Cutaneous barrier function is important for maintaining cutaneous conditions and plays significant roles in defending the skin against external stimuli such as allergens [15,16,17]. When the barrier is perturbed, multiple reactions occurred in the skin, such as the induction of cytokine cascade [18] and protease activation [19]. The barrier perturbation also causes a Th2-dominant immunophenotype of AD [20,21,22,23]. The cutaneous barrier function of AD subjects is impaired not only on the lesional skin, but also the non-lesional skin.

Furthermore, water in the stratum corneum (SC) maintains its softness and flexibility. If the water content in SC decreases, SC becomes hard and brittle, resulting in dry skin and skin itching easily occurring with nonspecific stimulation. Water in the epidermis is also required for metabolic processes. In different keratinocytes, enzymes depend on water content for their activity [24,25,26,27].

The permeability barrier localized in the SC consists of corneocytes and intercellular lipids. Corneocytes are composed of a cornified envelope, a keratin-filled interior, and are connected by corneodesmosomes. Filaggrin is a key component of the epidermal differentiation complex of the SC in the epidermal layer of human skin. It plays an important role in the barrier function of the skin and also forms the natural moisturizing factor (NMF) in the SC [28]. The keratohyalin granules in the granular layers are predominantly composed of profilaggrin [29]. As the cornified envelope forms, filaggrin molecules bind to and aggregate the keratin filaments, causing flattening of the cell to form the corneocyte. After aggregation of the keratin filaments, filaggrin is degraded to amino acids by a combination of bleomycin hydrolase, caspase 14 and calpain 1 [30,31]. Loss-of-function mutations in the filaggrin gene (FLG) predispose it to AD [32,33,34,35,36]. This down-regulation of the FLG gene might be related to the defective epidermal function [37,38,39,40,41,42].

The intercellular lipids that form lamellae structures are also crucial for maintaining the cutaneous permeability barrier and water holding function [43,44]. These lipids are organized in a predominant repeat distance of approximately 13 nm [45]. These lipids account for about 10% of the tissue mass of the SC, with a composition of approximately 50% lipid mass ceramides, 25% cholesterol, and 15% free fatty acids, with a small amount of phospholipid [46,47]. The ceramides in the SC are unusual and very diverse, with a high percentage of very-long-chain N-acyl fatty acids. There are around 25 ceramide subclasses identified [48]. Most of these ceramides have a sphingoid base and an acyl chain linked by an amide bond. The acyl chain is designated as non-hydroxy (N), α-hydroxy (A), ω-hydroxy (O), esterfied-hydroxy (EO) or β-hydroxy acyl (B). In human SC, the sphingoid bases of the ceramides are dihydrosphingosine (dS), sphingosine (S), phytosphingosine (P), 6-hydroxy-sphingosine (H) and 4.14 sphingadiene (SD). In addition to the variation in subclasses, their total chain length (sphingoid base + acyl chain) varies substantially between around 32 and 72 carbon atoms. The variation in the length of the sphingoid bases is much less than the variation in the acyl chain length [48,49,50]. The ceramides’ EO subclasses are defined as all subclasses with a very long ω-hydroxy acyl chain ester linked to a fatty acid chain with 18 carbon atoms, the majority of which is a linoleate [51]. The ceramides are essential for cutaneous barrier function, and the decreases in ceramide levels perturb this SC function. The papers about the clinical usefulness of preparations containing natural or synthetic ceramides for water retention and barrier functions were summarized in Kono’s report [52].

In the skin of patients with AD, decreased ceramide levels in the SC are observed, which correlate with cutaneous conditions [53,54,55,56,57,58]. Thus, the barrier perturbation and dry skin of patients with AD may be mainly attributable to a decrease in ceramides in the SC. Topically applied ceramides can improve both the permeability barrier and water holding functions when provided at sufficient concentrations [59]. Topically applied lipids do not only form an occlusive layer on the SC surface but also help to form lamellar structures in intercellular spaces. The efficacies of topically applied ceramides [60,61], pseudo-ceramide [62,63,64,65,66] and precursors of ceramide [67], have been reported for patients with AD, which are summarized in Table 1. The reports were searched for in PubMed with the key words, “atopic dermatitis” and “ceramide” or “pseudoceramimde”. To apply sufficient concentrations, a synthetic pseudo-ceramide, cetyl PG hydroxyethyl palmitamide (SLE66), was developed using molecular designs [68]. Further, it was shown that eucalyptus leaf extract (ELE) can enhance ceramide synthesis in the epidermis, which results in reinforcing SC functions [69]. Most reports shown in Table 1 indicate the efficacy or safety of ceramide or pseudo-ceramides on the skin symptoms of AD only. However, we have evaluated the efficacy of SLE66 and ELE on cutaneous conditions of AD with various points. This is the first review summarizing these studies. Table 2 indicates the points of analysis of SLE66 and ELE in AD.

## 2. Atopic Dermatitis and Ceramides

Ceramides are composed of long-chain fatty acids linked to sphingoid bases by amide bonds. In human skin, ceramides consist of four kinds of sphingoids and three kinds of fatty acids that result in a total of twelve distinct types of ceramides. Furthermore, each type of ceramide has various carbon numbers of fatty acids and ceramides in intercellular lipids that exist as a mixture of these ceramide species (ceramide profile) [104]. It has been reported that there are ceramide deficiencies in AD skin, even in non-lesional SC [53,54,55,56,57,105]. The analysis of ceramides in the SC has revealed that AD skin contains not only lower levels of ceramides but also that the ceramide profile is different from that of healthy skin, which correlates with SC functions [58]. The ceramide deficiency in AD skin is caused by lower levels of ceramides containing non-hydroxy fatty acids and 6-hydroxysphingosines (Cer[NH]), ceramides containing non-hydroxy fatty acids and phytosphingosines (Cer[NP]), ceramides containing ester-linked ω-hydroxy fatty acids and sphingosines (Cer[EOS]), ceramides containing ester-linked ω-hydroxy fatty acids and 6-hydroxysphingosines (Cer[EOH]) and ceramides containing ester-linked ω-hydroxy fatty acids and phytosphingosines (Cer[EOP]). Interestingly, the level of ceramides containing a-hydroxy fatty acids and sphingosines (Cer[AS]) is significantly higher in AD skin. In addition, the larger species of ceramides containing non-hydroxy fatty acids and sphingosines (Cer[NS]); ceramides containing non-hydroxy fatty acids and dihydrosphingosines (Cer[NDS]), Cer[NH], Cer[AS]; and ceramides containing a-hydroxy fatty acids and 6-hydroxysphingosines (Cer[AH]) in AD subjects tended to be expressed at lower levels, whereas the smaller species of Cer[NS], Cer[NDS] and Cer[AS] tended to be expressed at higher levels than in healthy individuals.

De novo synthesis of ceramide starts in the stratum spinosum [106] with the condensation of serine and palmitoyl-CoA to form 3-ketodihydrosphingosine by serine palmitoyl transferase [107] as the first step. The biosynthesized ceramides are converted to glucosylceramides and sphingomyelin and stored in lamellar bodies [108,109]. In the final step of cornification, these lipids are converted to ceramides again by beta-glucocerebrosidase or sphingomyelinase and form lipid lamellae oriented approximately parallel to the corneocytes at the interface between stratum granulosum and SC [110]. And some ceramides are degraded to sphingosine by ceramidase [111]. In AD skin, neither beta-glucocerebrosidase nor ceramidase activities were different from those in healthy skin [112]. However, sphingomyelin deacylase, which hydrolyzes sphingomyelin and glucosylceramide at the acyl site to yield sphingosylphosphorylcholine and glucosylsphingosine, respectively, instead of ceramide, is over-expressed in AD skin resulting in a decrease in ceramides [113,114,115]. And it is revealed that this sphingomyelin deacylase is an acid ceramidase-degrading β-subunit [116].

Ceramides form multi-layered lamellar structures between corneocytes with free fatty acids and cholesterol. Interestingly, cholesterol is also important for cutaneous barrier function [117,118]; however, no decrease in cholesterol was observed in AD skin [55].

Because the decrease in ceramides in the SC may cause a barrier deficiency in AD skin [53,54,55,56,57,58] and topically applied ceramides can improve both the permeability of the barrier and water holding functions when provided at sufficient concentrations [59], topically applied ceramides or precursors of ceramides in subjects with AD improve their skin symptoms [60,61].

## 3. The Efficacy of SLE66 and ELE on AD Skin Symptoms

However, using natural ceramides has many risks, such as its high cost and low stability. Thus, pseudo-ceramides have been developed to avoid these problems [66,68,119,120]. Synthetic pseudo-ceramide (cetyl PG hydroxyethyl palmitamide (SLE66)) was developed using molecular designs [68] based on the structure of Cer[NS] (Figure 1). Synthesized pseudo-ceramide derivatives are characterized by structures having both amide bonds and hydroxyl groups as hydrophilic units, as well as two long alkyl chains. From the analysis of alkyl chain properties, it is revealed that the presence of saturated straight alkyl chains whose structural characteristics are very similar to naturally occurring ceramide in the SC and the absence of unsaturation or methyl branching are necessary for forming multi-lipid bilayers in the SC. Analysis with differential scanning calorimetry and X-ray analysis revealed that this pseudo-ceramide constructed stable alpha form has a lamellar structure with fatty acids and cholesterol [121]. The topical application of this compound caused a significant recovery of the water holding capacity accompanied by an improvement in the scaling of dry skin induced by solvent treatment.

ELE-increased ceramide levels both in cultured human keratinocytes and in human SC. This increase in ceramide levels was caused by topically applying ELE to the SC [69]. It was shown that macrocarpal A is the key component that stimulates ceramide synthesis (Figure 2) [122]. It is revealed that macrocarpal A enhances ceramide synthesis with an increase in the mRNA expression of serine palmitoyltransferase, acid sphingomyelinase, neutral sphingomyelinase, glucosylceramide synthase and glucocerebrosidase in a dose-dependent manner.

The skin moisturizer containing SLE66 and ELE (P-Cer moisturizer) contains more than 3% of SLE and more than 0.9% of ELE [92,123]. Evaluations of the efficacy of P-Cer moisturizer on mild to moderate AD skin indicate that more than 60% of patients with AD significantly improved their skin symptoms accompanied by improvement in their SC functions [62,63,64,65]. Skin dryness, scaling, itching, and follicular papules were significantly improved [64]. These treatments indicated similar efficacies with the topical treatment of urea or heparinoid derivatives [62,63]. Interestingly, the 4 weeks of topically applied P-Cer moisturizer decreased the sensitivity against mite antigen, which was revealed by a patch test [63]. Furthermore, accompanying the improvement of SC functions, erythema and itchiness significantly decreased, which corelated with the improvement in skin dryness [124].

## 4. The P-Cer Moisturizer Improves Ceramide Profiles in AD Skin

The difference in ceramide levels in the SC of AD skin to healthy skin is not only due to the amount of each ceramide but also the total ceramide profile [58]. However, the topical application of the P-Cer moisturizer changed the AD skin ceramide profile to a healthy skin ceramide profile [92]. Four weeks of treatment with the P-Cer moisturizer significantly reduced the skin symptoms, such as erythema, desquamation, lichenification, excoriation, stinging, burning and itching. Based on this evaluation, as well as the improvement in skin symptoms, the cutaneous barrier and water holding function were significantly improved. Accompanying those improvements, the levels of Cer[NH] and Cer[NP] were increased and the levels of Cer[NS] and Cer[AS] were decreased. Also, the levels of the larger species of Cer[NS] were increased. These ceramide profiles are similar to a healthy skin phenotype. SLE66 penetrated the SC, and this penetrated SLE66 level was significantly correlated with the SC water content and the level and average carbon chain length of Cer[NS] [92]. Thus, the penetrated SLE66 may have contributed to these changes in the ceramide profile from an AD skin type to a healthy skin type. It is reported that the Cer[NP]/Cer[NS] ratio in the SC is correlated to epidermal differentiation [125]. Thus, it is presumable that the disrupted keratinization process resulted from cutaneous inflammation may cause the ceramide profile changes. And P-Cer moisturizer may reduce this cutaneous inflammation with the improvement in the cutaneous barrier and water holding function, switching the ceramide profile to a healthy skin phenotype by normalizing the epidermal keratinization.

## 5. The P-Cer Moisturizer Improves AD Even in the Summer Season

Patients with AD recognize that treatments with a moisturizer are beneficial for their dry skin during the dry and/or cold winter season. However, in the less dry summer season, many of them pay less attention to skin moisturizing, and many patients with AD stop applying moisturizers in the summer because of the high humidity and excessive sweating. In fact, accompanied by the increase in atmospheric temperature, skin dryness and scaling significantly improved without the application of moisturizers. However, the improvement in skin dryness in the P-Cer moisturizer users was significantly higher than in the non-moisturizer users, even in the summer season [93]. Furthermore, erythema and itchiness significantly improved, and skin hydration on the forearm increased significantly in the P-Cer moisturizer users, but not in the non-moisturizer users. Accompanied by these improvements, the QoL of patients with AD analyzed with the Skindex-16^®^ revealed significant improvements in emotions, functioning and global after 4 weeks of usage. Thus, the high humidity conditions in the summer may decrease skin dryness and scaling but not enough to prevent the skin symptoms of AD, such as erythema and itchiness, and the P-Cer moisturizer is effective for the skin care of patients with AD, even in the summer. Confirmation of the efficacy of P-Cer moisturizer in South and East Asia may be coincident with this result [89,96,126].

## 6. The P-Cer Moisturizer Improves Sweating

Many patients with AD have a significant decrease in sweat volume [127,128,129,130,131]. Since sweat regulates body temperature, decreased perspiration may cause the retention of body heat and it moisturizes the skin [132], controls the skin pH [133] and contains antimicrobial materials [134,135]. Thus, decreases in perspiration may be a crucial factor that worsens the skin symptoms of patients with AD.

Various factors might induce a decrease in perspiration in patients with AD, for example, the histamine produced in inflammation inhibits sweating [136,137]. Although acetylcholine levels are increased in the skin of patients with AD [138], the expression of acetylcholine receptors is reduced [139]. The decreased number of tight junctions in the sweat glands of patients with AD leads to sweat leakage [140,141]. Furthermore, the occlusion of sweat pores caused by hyper-keratinization in AD skin might also be a crucial factor that prevents sweating [142,143]. Davis and Lawler confirmed that the removal of superficial corneocytes by tape-stripping recovered the perspiration of subjects with AD [144]. Papa and Kligman reported that decreasing the occlusion of sweat pores by improving parakeratosis might be effective for the recovery of perspiration [136]. Thus, improving SC functions may help to improve perspiration. It was reported that ceramides modulate acetylcholine receptor levels [144], and activation of the sphingosine-1-phosphate receptor induces the synthesis of claudin-3 [145,146].

We have evaluated the efficacy of P-Cer moisturizers on the sweating function in adult patients with AD in a double-blind, randomized, controlled left–right comparison clinical trial. A P-Cer moisturizer was topically applied on the cubital fossa of one arm of each patient with AD and a moisturizer without P-Cer was applied topically on the cubital fossa of the other arm of each patient. Following application twice a day for 4 weeks, the skin conditions and sweating ability, measured as the response to acetylcholine stimulation, were evaluated. Both moisturizers (with or without P-Cer) improved the visually evaluated skin symptoms and skin hydration. However, only the P-Cer moisturizer significantly improved the cutaneous barrier function and significantly increased the ceramide level in the SC. Accompanied by these improvements, the sweating volume and a shortened latency time for sweating, an indicator of sweating ability, were observed. Although the moisturizer without P-Cer also improved the visually evaluated skin symptoms and skin hydration, it did not improve the sweating function. Thus, SLE66 and/or the increase in ceramides with ELE may help recover the sweat function of patients with AD [90]. Mizukawa’s group reported that dupilumab restored the sweating disturbance in addition to improving AD symptoms [147]. Our results may indicate that treatment with P-Cer moisturizer has the ability to improve the sweating function of patients with AD as a clinical treatment.

## 7. Filaggrin Mutations and the P-Cer Moisturizer

Filaggrin is associated with keratin intermediate filaments that work in the cutaneous barrier function, and its metabolites work as NMFs [148,149]. Loss-of-function FLG mutations are genetic risk factors for AD [150]. Decreases in filaggrin function may induce weakening of corneocytes and decrease the level of NMF [151,152]. Although decreases in intercellular lipids and in filaggrin are crucial factors for AD skin symptoms, the lipid metabolism in AD might be independent of the FLG genotype [153], and AD with a normal FLG genotype has aberrant changes in the expression of various enzymes involved in the metabolism and synthesis of lipids. Furthermore, the frequency of FLG mutations was not significantly different between the AD and non-AD groups in a cohort of children from Ishigaki Island [154]. Filaggrin is one of the sources of natural moisturizing factors in the SC, but sweat also supplies moisturizing factors [132,155]. The lactate and potassium in the sweat are essential for skin moisturizing [156]. Thus, in a warm environment such as in Japan, sweating in daily life may help to moisturize skin, even in those with FLG mutations. Thus, skin care for moisturizing and reinforcing the barrier function of skin may be effective to prevent the induction of skin symptoms of AD, even in patients with FLG mutations. The effect of FLG mutations on the efficacy of the P-Cer moisturizer in patients with AD was evaluated at Oita University [87]. Of the 39 patients with AD studied, 7 had at least one of the FLG loss-of-function mutations that have been reported in Japan [157,158,159]. In that evaluation, FLG mutations made no significant difference in maintaining the remission phase between AD exacerbation (−) and AD exacerbation, suggesting that there might be no correlation between the existence of FLG mutations and the efficacy of P-Cer skin care.

## 8. The Ceramide Profile Affects the Efficacy of the P-Cer Moisturizer to Maintain the Remission Phase

Although FLG mutations may not affect the efficacy of P-Cer skin care, there are some patients with AD who have difficulty in controlling their skin symptoms following topical corticosteroid treatments. Although there were no differences in skin severity, analysis of the SC ceramide profiles revealed that some patients with AD had a ceramide profile that was similar to the healthy skin phenotype and some patients with AD had a ceramide profile with a lower ratio of Cer[NP]/Cer[NS] and a lower carbon number of acyl chains of Cer[NS]. Patients with AD with a heathy ceramide profile could maintain the remission phase; however, patients with AD with an abnormal ceramide profile did not improve, even after topical corticosteroid treatments [87]. Generally, the relief of AD skin symptoms causes an improvement in ceramide profiles. The Cer[NP]/Cer[NS] ratio in the SC might be a potential marker for skin properties and epidermal differentiation, and a low ratio may indicate immature differentiation of the SC [125]. Thus, this evaluation may indicate that abnormal ceramide profiles, which may be caused by perturbed keratinization, had even fewer skin symptoms following medical treatment, and even inflammation might be equally relieved, as there were no significant differences in TARC values. Further, the P-Cer moisturizer was not effective in patients with AD with keratinocytes that are not normalized with respect to their turnover, even with corticosteroid treatments.

## 9. The P-Cer Moisturizer and Skin Cleansing

Regarding skin care for patients with AD, skin cleansing also affects their skin conditions [160,161]. As mentioned above, the P-Cer moisturizer is effective for AD care, even in higher temperature conditions and also in South and East Asia [96,126]. Concerning the evaluation of the efficacy of P-Cer moisturizer in patients with mild AD in Thailand, this moisturizer indicates a significant improvement both in the SCORing Atopic Dermatitis (SCORAD) index and subjectively using Patient-Oriented Eczema Measure (POEM) scores. In addition to this improvement in skin symptoms by application of the test lotion, the use of a mild body cleanser [162] induced a significantly higher improvement in both evaluations [89]. Effective cleansing and retaining skin moisture are conflicting propositions and have been called the ‘cleansing paradox’. This mild cleanser mainly formulated with sodium laureth sulphate supplemented with sodium laureth carboxylate and lauryl glucoside has a high sebum cleansing ability and also caused decreases in scaling and erythema in subjects with dry skin in a controlled usage study [162].

This result may suggest that P-Cer moisturizer can partly compensate for the effect of body cleansing, but it is not sufficient. Thus, both mild and suitable body cleansing and P-Cer moisturizing may be effective in maintaining inflammation-relieved skin.

## 10. Summary

Moisturizing skin is quite important for the treatment of AD. There are some reports of many patients with AD experiencing improved skin conditions in the summer compared to the winter. Filaggrin is an important material for cutaneous moisture content, and loss-of-function FLG mutations are genetic risk factors for AD; however, in a cohort study of children from Ishigaki Island, a decrease in filaggrin might have had less of an effect on the condition within a humid environment [154]. In contrast, topical treatment with P-Cer moisturizer is effective even in the summer and in South and East Asia. Also, there is no correlation between the existence of FLG mutations and the efficacy of P-Cer moisturizer.

Ceramides, which are crucial for SC function, are decreased in the SC of patients with AD, resulting in induced cutaneous barrier perturbation and skin dryness. Cutaneous barrier perturbation and skin dryness are major factors of atopic dermatitis. These SC disfunctions induced various skin symptoms such as severe immune responses, itching, inflammation, sweating abnormities, and abnormal epidermal turnover. And this abnormal turnover induces a decrease in ceramide contents and also an abnormal ceramide profile, which has less efficient SC function. Thus, controlling epidermal proliferation and differentiation might be also effective. For example, vitamin D enhances the differentiation of keratinocytes and reinforces the cutaneous barrier function [163]. Vitamin D has the potential to modulate allergies, the epidermal barrier function, immune dysregulation, and bacterial defense [164,165]. Thus, these materials may be useful for AD skin care to increase the barrier and water holding function.

The P-Cer moisturizer that contains synthetic pseudo-ceramide and ELE with sufficient concentrations may compensate this decreased ceramide, resulting in improving the cutaneous barrier and water holding functions of patients with AD. By reinforcing these functions of SC, skin itchiness, erythema, and inflammation are relieved, and epidermal turnover is normalized, resulting in improved sweating function and ceramide metabolism. This ceramide metabolism increases ceramide contents with healthy profiles in the SC. As shown here, P-Cer moisturizer is an excellent and useful material for treating AD skin. 

Recently, the efficacy of drugs that contain P-Cer has been confirmed. A cream containing heparinoid and P-Cer forms lamellar structures that result in inhibiting allergen penetration with an improvement in SC functions. This cream is effective in maintaining the AD remission period [91]. The cream with steroids and P-Cer, which also forms a lamellar structure, not only demonstrates anti-inflammatory effects but also significantly improves SC functions [88].

In this review, I summarize the studies which have evaluated the efficacy of P-Cer moisturizer in cutaneous conditions of AD with various points. Of course, the formulations of the moisturizers are very important for these efficacies, but the similar formulation without SLE66 or ELE was less effective in patients with AD. It is presumed that natural ceramides with sufficient concentrations may indicate similar efficacies, but we have not compared the efficacy of P-Cer with that of natural ceramides. Further studies with natural ceramides or pseudo-ceramides may discover more effective treatments for AD.

## Figures and Tables

**Figure 1 jcm-13-01749-f001:**
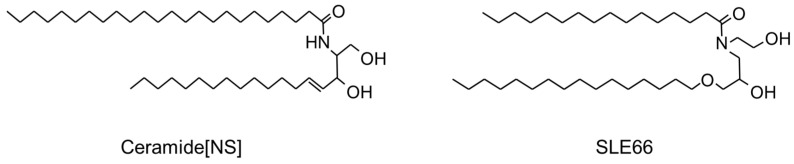
Structure of synthesized pseudo-ceramide (SLE66).

**Figure 2 jcm-13-01749-f002:**
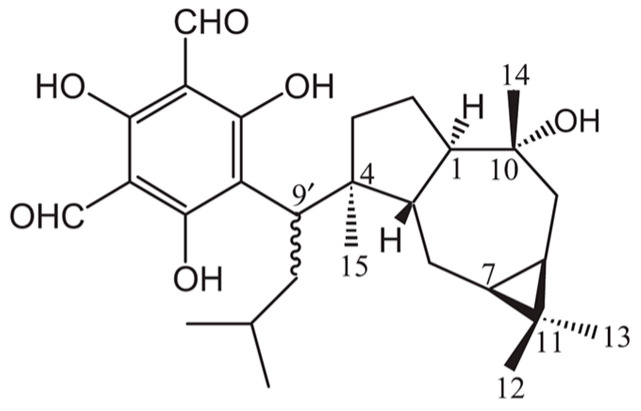
Structure of Macrocarpal A.

**Table 1 jcm-13-01749-t001:** A list of manuscripts about the efficacy of ceramides or pseudo-ceramides written in English.

Authors	Journal	Ceramides or Pseudo-Ceramides	Emollients /Pseudo-Ceramides	Ref.
Spada, F.; et al.	Dermatol Ther. (2021)	ceramide	Ceradan Cream	[70]
Zirwas, M.J; et al.	J Drugs Dermatol. (2017)	ceramide	CeraVe^®^	[71]
Koppes, S.A.; et al.	Acta Derm Venereol. (2016)	ceramide	Cer[EOS], [NP], [AP]	[72]
Somjorn, P.; et al.	Asian Pac J Allergy Immunol. (2021)	ceramide	EpiCeram	[73]
Kircik, L.H.; et al.	J Clin Aesthet Dermatol. (2011)	ceramide	EpiCeram	[74]
Kircik, L.H.; et al.	J Clin Aesthet Dermatol. (2011)	ceramide	EpiCeram	[75]
Draelos, Z.D.	J Cosmet Dermatol. (2011)	ceramide	EpiCeram	[76]
Miller, D.W.	J Drugs Dermatol. (2011)	ceramide	EpiCeram	[77]
Sugarman, J.L.; et al	J Drugs Dermatol. (2009)	ceramide	EpiCeram	[78]
Puviani, M.; et al.	Minerva Pediatr. (2014)	ceramide	Cer[NP]	[79]
Park, K.Y.; et al.,	J Korean Med Sci. (2010)	ceramide	TriCeram	[80]
Chamlin, S.L.; et al.	Arch Dermatol. (2001)	ceramide	TriCeram	[61]
Chamlin, S.L.; et al.	J Am Acad Dermatol. (2002)	ceramide	TriCeram	[81]
Berardesca, E.; et al.	Contact Dermatitis. (2001)	ceramide	Cer[NP]	[60]
Gupta, S.; et al.	Pediatr Dermatol. (2023)	ceramide	no detail data	[82]
Tey, H.-L.; et al.	Skinmed. (2017)	ceramide	no detail data	[83]
Yang, Q.; et al.	Dermatol Ther. (2019)	ceramide	no detail data	[84]
Lynde, C.W.; et al.	Cutis. (2014)	ceramide	no detail data	[85]
Takada, M.; et al.	Int J Mol Sci. (2022)	pseudo-ceramide	SLE66 + ELE	[86]
Sho, Y.; et al.	J Invest Dermatol. (2022)	pseudo-ceramide	SLE66 + ELE	[87]
Okoshi, K.; et al.	Dermatol Ther (Heidelb). (2022)	pseudo-ceramide	SLE66 + ELE	[88]
Nojiri, H.; et al.	J Cosmet Dermatol. (2022)	pseudo-ceramide	SLE66 + ELE	[89]
Shindo, S.; et al.	J Cosmet Dermatol. (2022)	pseudo-ceramide	SLE66 + ELE	[90]
Matsuoka, M.; et al.	Clin Cosmet Investig Dermatol. (2021)	pseudo-ceramide	SLE66 + ELE	[91]
Ishida, K.; et al.	J Invest Dermatol. (2020)	pseudo-ceramide	SLE66 + ELE	[92]
Mori, K.; et al.	J Cosmet Dermatol. (2019)	pseudo-ceramide	SLE66 + ELE	[93]
Hon, K.L.; et al.	Hong Kong Med J. (2011)	pseudo-ceramide	SLE66 + ELE	[94]
Hon, K.L.; et al.	Curr Pediatr Rev. (2018)	pseudo-ceramide	SLE66 + ELE	[95]
Seghers, A.C.; et al.	Dermatol Ther (Heidelb). (2014)	pseudo-ceramide	SLE66 + ELE	[96]
Matsuki, H.; et al.	Exog. Dermatol. (2004)	pseudo-ceramide	SLE66 + ELE	[65]
Ma, L.; et al.	Adv Ther. (2017)	pseudo-ceramide	Hydroxypropyl Bispalmitamide MEA	[97]
Ho, Y.V.P.; et al.	Dermatol Ther (Heidelb). (2020)	pseudo-ceramide	Hydroxypropyl Bispalmitamide MEA	[98]
Koh, M.J.-A.; et al.	Dermatol Ther (Heidelb). (2017)	pseudo-ceramide	Hydroxypropyl Bispalmitamide MEA	[99]
Chang, A.S.L.; et al.	Geriatr Nurs. (2018)	pseudo-ceramide	Hydroxypropyl Bispalmitamide MEA	[100]
Draelos, Z.D.; et al.	J Clin Aesthet Dermatol. (2018)	pseudo-ceramide	PC104	[101]
Na, J.-I.; et al.	J Dermatolog Treat. (2010)	pseudo-ceramide	PC104	[102]
Hon, K.L.; et al.	Drugs R D. (2013)	ceramide-precursor		[103]
Simpson, E.; et al.	J Dermatolog Treat. (2013)	ceramide-precursor		[67]

**Table 2 jcm-13-01749-t002:** The points of efficacy of SLE66 and ELE in AD.

1. Improvement of skin symptoms
2. Changes in ceramide profiles
3. Effective in the summer season
4. Improvement of sweating function
5. Correlation with filaggrin mutations
6. The potential biomarker to maintain the remission phase
7. Correlation with skin cleansing

## Data Availability

Not applicable.

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
