# Peer review of "Efficacy of Topical Application of a Skin Moisturizer Containing Pseudo-Ceramide and a Eucalyptus Leaf Extract on Atopic Dermatitis: A Review"

_jcm, 2024, doi:10.3390/jcm13061749_

Round 1

Reviewer 1 Report

Comments and Suggestions for Authors

1. the Introduction is too short for a review-type article. Please expand on the following ideas:

            - how management strategies have evolved over time in terms of treating atopic dermatitis

            - explain the biological mechanisms by which ceramides and pseudo-ceramides restore the skin barrier

            - discuss the development and innovation of synthetic pseudo-ceramides like SLE66

2. Include specific study findings or meta-analysis results that support the efficacy of ceramides, pseudo-ceramides, and their precursors in AD treatment

3. authors must consider to clearly state the unique contribution of this review. For example, if this is the first review to analyze the combined effects of SLE66 and eucalyptus leaf extract - highlight this.

4. authors should consider expanding on the biological role of ceramides in maintaining skin barrier integrity and how their reduction exacerbates AD

5. briefly summarize the current state of research on AD treatment and identify any gaps or limitations that your review aims to address

6. Chemical names should be formatted consistently throughout the document - there are some errors and typos

7. The spacing and formatting of references vary throughout the text - please revise

8. Provide an explanation of how FLG mutations influence treatment outcomes with the P-Cer moisturizer

Comments on the Quality of English Language

- The spacing and formatting of references vary throughout the text - please revise

- Chemical names should be formatted consistently throughout the document - there are some errors and typos

Author Response

Thank you for your kind reviewing of our manuscript entitled “Efficacy of topical application of a skin moisturizer containing pseudo-ceramide and a eucalyptus leaf extract on atopic dermatitis: A Review”. In response to the reviewer’s criticism, I have extensively modified the manuscript. As requested by you, the followings are amendments which we made in the revised manuscript and answers to the questions from the reviewer.

*****************************************************************************************

  1. the Introduction is too short for a review-type article. Please expand on the following ideas:

  - how management strategies have evolved over time in terms of treating atopic dermatitis

  - explain the biological mechanisms by which ceramides and pseudo-ceramides restore the skin barrier

  - discuss the development and innovation of synthetic pseudo-ceramides like SLE66

Answer

In light of the reviewer’s comments, I have expanded the introduction by adding some sentences about treatments for atopic dermatitis, research about treatments with ceramides or pseudoceramides including the mechanism. About the SLR66 development, I add sentences in “3. The efficacy of SLE66 and a eucalyptus leaf extract on AD skin symptoms”.

*****************************************************************************************

  1. Include specific study findings or meta-analysis results that support the efficacy of ceramides, pseudo-ceramides, and their precursors in AD treatment

Answer

I summarizer the research about the efficacy of ceramides, pseudo-ceramides, and their precursors in AD treatment in Table 1.

*****************************************************************************************

  1. authors must consider to clearly state the unique contribution of this review. For example, if this is the first review to analyze the combined effects of SLE66 and eucalyptus leaf extract - highlight this.

Answer

In light of the reviewer’s comments, I highlighted the contribution of this review on the endo of introduction with Table 2.

*****************************************************************************************

  1. authors should consider expanding on the biological role of ceramides in maintaining skin barrier integrity and how their reduction exacerbates AD

Answer

I added some sentences about the mechanism of the efficacy of P-Cer in section “1. Introduction, “3. The efficacy of SLE66 and ELE on AD skin symptoms”, and “10. Summary”.

*****************************************************************************************

  1. briefly summarize the current state of research on AD treatment and identify any gaps or limitations that your review aims to address

Answer

I added some sentences at the end of summary.

*****************************************************************************************

  1. Chemical names should be formatted consistently throughout the document - there are some errors and typos

Answer

I corrected the format of chemical names.

*****************************************************************************************

  1. The spacing and formatting of references vary throughout the text - please revise

Answer

I corrected the format of references.

*****************************************************************************************

  1. Provide an explanation of how FLG mutations influence treatment outcomes with the P-Cer moisturizer

Answer

I added some sentences in the section, “7. Filaggrin mutations and the P-Cer moisturizer”.

*****************************************************************************************

Comments on the Quality of English Language

- The spacing and formatting of references vary throughout the text - please revise

Answer

I corrected the format of references.

*****************************************************************************************

- Chemical names should be formatted consistently throughout the document - there are some errors and typos

Answer

I corrected the format of chemical names.

Reviewer 2 Report

Comments and Suggestions for Authors

The author higlights how it is important in AD patients increase the ceramides in the skin, in order to maintain the cutaneous barrier. Specifically the author report the use of a pseudo-ceramide (cetyl PG hydroxyethyl palmitamide) which has a structure developed via molecular designs, and a eucalyptus leaf extract enhance  ceramide synthesis in the epidermis. Despite the article is of interese some changes are needed:

- Lines 23-27: please ass more sentences about atopic dermatitis, reporting the main clinical aspects of the disease and its pathogenesis

- Add some sentence about the cutaneous barrier and the role of Vitamin D. Indeed, you may report that in AD the alteration of the cutaneous barrier plays a pivotal role in its pathogenesis, as well as the interaction with Vitamin D that (with some difference between sun exposed and non sun exposed areas) may reduce the inflammation and the keratinocyte proliferation  and barrier alteration. Accordingly please add this interesting reference:  doi: 10.1007/s12032-014-0451-4; PMID: 25516505

- Please adda a Table summarizing some aspects of your review

- Since you state that a P-Cer moisturizer that contains synthetic pseudo-ceramide and a eucalyptus leaf extract mY improve the SC functions of AD patients accompanied by normalizing the ceramide pro-file in the SC, add some sentence about the mechanism of action.

Author Response

Thank you for your kind reviewing of our manuscript entitled “Efficacy of topical application of a skin moisturizer containing pseudo-ceramide and a eucalyptus leaf extract on atopic dermatitis: A Review”. In response to the reviewer’s criticism, I have extensively modified the manuscript. As requested by you, the followings are amendments which we made in the revised manuscript and answers to the questions from the reviewer.

*****************************************************************************************

The author higlights how it is important in AD patients increase the ceramides in the skin, in order to maintain the cutaneous barrier. Specifically the author report the use of a pseudo-ceramide (cetyl PG hydroxyethyl palmitamide) which has a structure developed via molecular designs, and a eucalyptus leaf extract enhance  ceramide synthesis in the epidermis. Despite the article is of interese some changes are needed:

- Lines 23-27: please ass more sentences about atopic dermatitis, reporting the main clinical aspects of the disease and its pathogenesis

Answer

In light of the reviewer’s comments, I have expanded the introduction by adding some sentences about treatments for atopic dermatitis, research about treatments with ceramides or pseudoceramides including the mechanism. About the SLR66 development, I add sentences in “3. The efficacy of SLE66 and a eucalyptus leaf extract on AD skin symptoms”.

*****************************************************************************************

- Add some sentence about the cutaneous barrier and the role of Vitamin D. Indeed, you may report that in AD the alteration of the cutaneous barrier plays a pivotal role in its pathogenesis, as well as the interaction with Vitamin D that (with some difference between sun exposed and non sun exposed areas) may reduce the inflammation and the keratinocyte proliferation  and barrier alteration. Accordingly please add this interesting reference:  doi: 10.1007/s12032-014-0451-4; PMID: 25516505

Answer

In light of the reviewer’s comments, I added some sentences in section “10. Summary” and references including the recommended paper.

*****************************************************************************************

- Please adda a Table summarizing some aspects of your review

Answer

In light of the reviewer’s comments, I added a table in introduction section.

*****************************************************************************************

- Since you state that a P-Cer moisturizer that contains synthetic pseudo-ceramide and a eucalyptus leaf extract mY improve the SC functions of AD patients accompanied by normalizing the ceramide pro-file in the SC, add some sentence about the mechanism of action.

Answer

I added some sentences in section “4. The P-Cer moisturizer improves ceramide profiles in AD skin”.

*****************************************************************************************
